# Modeling cannabinoids from a large-scale sample of *Cannabis sativa* chemotypes

**Daniela Vergara**[1]*, **Reggie Gaudino**[2], **Thomas Blank**[2], **Brian Keegan**[3]*

**1** Department of Ecology and Evolutionary Biology, University of Colorado, Boulder, Colorado, United States of America, **2** Front Range Biosciences, Lafayette, CO, United States of America, **3** Department of Information Science, University of Colorado Boulder, Boulder, Colorado, United States of America

* daniela.vergara@colorado.edu (DV); brian.keegan@colorado.edu (BK)

## Abstract

The widespread legalization of *Cannabis* has opened the industry to using contemporary analytical techniques for chemotype analysis. Chemotypic data has been collected on a large variety of oil profiles inherent to the cultivars that are commercially available. The unknown gene regulation and pharmacokinetics of dozens of cannabinoids offer opportunities of high interest in pharmacology research. Retailers in many medical and recreational jurisdictions are typically required to report chemical concentrations of at least some cannabinoids. Commercial cannabis laboratories have collected large chemotype datasets of diverse *Cannabis* cultivars. In this work a data set of 17,600 cultivars tested by Steep Hill Inc., is examined using machine learning techniques to interpolate missing chemotype observations and cluster cultivars into groups based on chemotype similarity. The results indicate cultivars cluster based on their chemotypes, and that some imputation methods work better than others at grouping these cultivars based on chemotypic identity. Due to the missing data and to the low signal to noise ratio for some less common cannabinoids, their behavior could not be accurately predicted. These findings have implications for characterizing complex interactions in cannabinoid biosynthesis and improving phenotypical classification of *Cannabis* cultivars.

## Introduction

Dozens of countries worldwide have legalized *Cannabis* for medicinal and recreational use. As of this writing, nine states and the District of Columbia legalized both the medical and recreational use of *Cannabis*, while 29 other states legalized its medical consumption. *Cannabis* legalization has opened the doors to a new—and profitable—worldwide industry, with significant potential for medical and agricultural research. Regulatory requirements have brought state-of-the-art analytical instrumentation into labs to ensure a proper analysis of the plant which contains hundreds of chemical moieties including cannabinoids.

*Cannabis* is an angiosperm (flowering) plant, which uniquely produces the cannabinoid compounds that drive this growing market. Some of these cannabinoids can interact with the human endocannabinoid system, and have demonstrated psychoactive effects and medicinal

**Funding:** This research was supported by donations to the Agricultural Genomics Foundation, and is part of the joint research agreement between the University of Colorado Boulder and Steep Hill Inc. The company Steep Hill Inc. provided support in the form of salaries for authors R.G. and T.B., but did not have any additional role in the study design, and analysis, decision to publish, or preparation of the manuscript. The specific roles of these authors are articulated in the 'author contributions' section."

**Competing interests:** D.V. is the founder and president of the non-profit organization Agricultural Genomics Foundation, and the sole owner of CGRI, LLC. R.G., and T.B. are employees of Front Range Biosciences and previously of Steep Hill, Inc.

potential [1] for example as COX-2 inhibitors [2]. The plant produces these compounds in the acidic form, and the acids are reduced into the neutral form, typically by heating. Two of the most widely known cannabinoids, Δ-9-tetrahydrocannabinolic acid (THCA) and cannabidiolic acid (CBDA), are converted respectively to their neutral forms Δ-9 tetrahydrocannabinol (THC) and cannabidiol (CBD) under heating. THC is well-known for its psychoactive effects [1].

The biochemical pathway responsible for the production of these compounds is complex and involves multiple genes [3]. The enzymes responsible for the production of CBDA and THCA—CBDA and THCA synthases (CBDAS or THCAS), respectively—act on the same precursor molecule: Cannabigerolic acid (CBGA) [4], along with a third poorly studied synthase Cannabichromenic acid synthase (CBCAS), the producer of Cannabichomenic acid (CBCA). These three synthases are found at the final stage of the biochemical pathway. They are codified by genes which are similar in their genetic sequence [3] and are close in proximity to each other [5–7]. These complex pathways complicate efforts to predict or design fundamental phenotypic characteristics like THCA production in a given strain.

Other minor cannabinoids include Cannabinol (CBN), Δ-9-tetrahydrocannabivarin carboxylic acid (THCVA), and Cannabidivarinic Acid (CBDVA). CBN is a product that accumulates with the breakdown of THC [8–10]. Both THCVA and CBDVA are produced in very low quantities by most *Cannabis* cultivars, and likely for this reason, they have not been studied widely to date. Some studies suggest that both THCVA and CBDVA are produced by a variant of the well-known cannabinoid pathway [11, 12] that yields to the production of homologous compounds such as cannabigerovarinic acid (CBGVA) which is the precursor molecule used by THCVA and CBDVA synthases [11, 12]. THCV the decarboxylated neutral of THCVA is the only other cannabinoid that is known to produce psychoactive effects along with THC [13], and CBDV may have possible medicinal properties as an anticonvulsant [14].

The number of species that compose this genus *Cannabis* is debated among scientists [15, 16]. The phenotypic and chemotypic characteristics of these different groupings—most frequently labeled as "indica" and "sativa" are an important factor for the plant's classification and use. "Sativa" type plants are reputed to produce high THCA and low CBDA, are described as having uplifting and stimulating psychoactive effects after consumption, and are utilized to treat depression, low energy, headaches, nausea, and loss of appetite [17]. Plants labeled as "Indica" are reputed to have relaxing and more sedative effects perhaps partly due to higher levels of CBDA [17, 18]. Indica labeled plants are typically used to treat depression, anxiety, insomnia, pain, inflammation, muscle spasms, epilepsy, and glaucoma [17]. The crosses between "sativa" and "indica" plants are referred to as "hybrids" and have variable chemotypes usually intermediate to the parents [18]. It follows that these distinctions are of critical importance for medical patients.

The taxonomic based naming convention from the *Cannabis* industry is flawed, varieties and groupings are often mislabeled, and their attempt at nomenclature is inconsistent with biological relatedness or therapeutic use[18–20]. This labeling problem was also previously limited by the narrow representations of cannabis cultivars approved by the US government for research in the past [21, 22].

This study examines a large sample of cultivars available in medical and recreational markets within the United States. We make two contributions: identifying latent clusters of cultivars based on similarities in their chemotypes and evaluating the accuracy of different methods for imputing missing chemotype data. There are coherent and interpretable clusters of cultivars reflecting product demand in the rapidly-changing market. However, no single imputation method is able to reliably predict all missing chemotype data, but some imputation methods do have very high accuracy for specific cannabinoids.

## Methods

### Data collection

Cannabinoid concentration profiles (chemotypes) were generated by Steep Hill, Inc following the companies published measurement protocol [20]. Briefly, data collection was performed using high performance liquid chromatography (HPLC) with Agilent (1260 Infinity, Santa Clara, CA) and Shimadzu (Prominence HPLC, Columbia, MD) equipment with 400–6000 mg of sample. The data analysis included only data from *Cannabis* flowers.

We combined the acidic forms of the cannabinoids and their neutral forms using the same conversion rate of 0.88 which was also used by Vergara et al. (2017). The combination of the acidic and neutral forms provides a more honest analysis given that these flowers could have been collected at different points in time and therefore their ratio of acidic to neutral forms of the cannabinoids could differ. We analyzed seven cannabinoids, four which are part of the same biochemical pathway [7]: CBG, CBD, THC, CBC and the breakdown product of THC, Cannabinol (CBN). We also included for some of our analyses two other cannabinoids whose biochemical pathways is still unknown: Tetrahydrocannabivarin (THCV) and Cannabidivarin (CBDV).

### Data distribution

The raw data provided by Steep Hill, Inc. consisted of 13 different files and is found in the dryad repository (https://doi.org/10.5061/dryad.sxksn0314). These data were taken from a combination of regulatory compliance testing, internal customer research and development, and data collected as part of studies directed to chemical or genetic characterization over a period of eight years. Therefore, these data represent a mixture of types that were combined and subsequently filtered based on depth of information. We used custom Python scripts using the pandas library to clean these raw files, which resulted in a single file with 17,600 entries, with samples from four different US states: 2 from Alaska; 16,418 from California; 936 from Colorado; and 244 from Washington. California was the only state that included samples from eight years (2011–2018), and the only year that included three states (California, Colorado, and Washington) was 2014.

### Missing data and imputation

Few of these cultivars ($N = 153$) have information about all of the major cannabinoids summarized in Table 1 with the features of the $N = 17,600$ unique cultivars in the data set. Given the overwhelming prevalence of cases missing at least one observation of a cannabinoid (Table 1,

**Table 1. Cannabinoid distribution for 17,612 unique cultivars.**

| Cannabinoid | Count | Missing | Mean | S.D. | Max. |
|---|---|---|---|---|---|
| Cannabigerol (CBG) | 12,270 | 30.28% | 0.50% | 0.47% | 11.58% |
| Δ9-tetrahydrocannabinol (THC) | 17,346 | 1.44% | 14.03% | 6.45% | 33.95% |
| Cannabidiol (CBD) | 10,948 | 37.80% | 1.68% | 3.91% | 25.80% |
| Cannabichromene (CBC) | 6,487 | 63.14% | 0.120% | 0.128% | 1.79% |
| Cannabinol (CBN) | 6,675 | 62.07% | 0.067% | 0.18% | 2.95% |
| Tetrahydrocannabivarin (THCV) | 3,311 | 81.19% | 0.198% | 0.412% | 6.62% |
| Cannabidivarin (CBDV) | 597 | 96.60% | 0.069% | 0.11% | 0.83% |

For each cannabinoid (column 1), we report the number of cases (column 2), the percentage missing (column 3), the average concentration (column 4), standard deviation of concentrations (column 5), and maximum observed value (column 8).

S1 Fig in S1 File), dropping these cases from the analysis would greatly reduce statistical power as well as introduce significant sampling biases. We follow statistical best practices about reporting missing data by reporting on the completeness of the data, detail our approaches for handling missing data, and explore the features of the missing data.

These data could be missing for a variety of reasons. Mechanisms of missingness include "missing completely at random" (MCAR), "missing at random" (MAR), and "missing not at random" (MNAR) [23]. Some clients may elect not to test for more cannabinoids than regulations require (typically only THCA and THC) while others test for complete profiles for research, product development, or marketing purposes. The complex regulation of cannabinoid synthesis [3] and the significant genetic diversity in *Cannabis* [3, 19, 20] complicate efforts to model the relationships among cannabinoids. Methods like flux balance analysis could be used to augment the data, however the required constraints, objective functions, and metabolic pathways have not yet been well-characterized in *Cannabis*.

We instead employ imputation-based methods to estimate the missing data values. Imputation specifically refers to a class of methods for estimating missing values in a data set. The central intuition behind imputation is that information like **1)** other cannabinoids in the observation, **2)** the distribution of observed cannabinoids across samples, and **3)** similarities to other observations with complete data can guide estimation of the missing values. We employ four distinct imputation methods—iterative, multiple, k-neighbors, and soft—to estimate the missing values in our dataset using the "fancyimpute" package in Python (https://github.com/iskandr/fancyimpute).The use of these imputation methods are theoretically defensible because many of these cannabinoids are part of the same biochemical pathways and thus their prevalence and concentration are strongly correlated rather than independent [4]. For example, CBGA is the precursor molecule to which THCA, CBDA, and CBCA synthases act on [3, 4], so we expect that the abundance of THCA, CBDA, and CBCA could be used to estimate the abundance of CBGA. We evaluate the performance of these imputation methods using several approaches: examining the covariance among cannabinoids, the deviance of the imputed distributions from the non-missing data, the performance of different classes of regression models, and finally visualizing the overlap of the imputed values with existing clusters of data using dimensionality reduction techniques.

### Dimensionality reduction and clustering

Dimensionality reduction are methods for transforming high-dimensional data like our seven cannabinoid observations per cultivar into a lower dimensional representation necessary for visualization. Dimensionality reduction can be a powerful tool for classification, visualization, and compression of high-dimensional data, but have traditionally been limited to linear techniques such as principal components analysis (PCA) or factor analysis [24]. However, many kinds of real-world data, like the cannabinoid chemotypes in the present study, have complex and nonlinear relationships that require alternative dimensionality reduction techniques. Investigations included the use of two recently-developed and non-linear dimensionality reduction techniques, t-distributed stochastic neighbor embedding (t-SNE) [25] and uniform manifold approximation and projection (UMAP) [26] to project the seven variables into two dimensions for the purposes of visualization and clustering. Unlike PCA, the resulting X and Y axes in t-SNE and UMAP do not have substantive statistical interpretations but their spatial proximity and shapes of clusters captures similarity that can be more clearly viewed in human visual pattern recognition of a low dimensional graph: closer points are more similar than more distant points. These non-linear projections are transductive (making sense from the observations themselves) rather than inductive (premises provide general rules), so

withholding a subset of the data (*e.g.*, the imputed values) will return a different projection than using the full set. Dimensionality reduction techniques were applied to the data using the scikit-learn and umap libraries in Python [26].

**Statistical modeling.**   To evaluate the performance of the imputation methods, we treated each cannabinoid as a dependent variable in a linear equation with the other cannabinoids as explanatory variables. This allows us to compare several different regression methods to estimate the fit of the model to the data. Regression models are drawn from the traditional ordinary least square linear regression as well as more advanced k-neighbors and support vector regression, which are designed for modeling non-linear relationships. Models are then fitted using data from each of the four different imputation methods. The performance of each model is evaluated using the coefficient of determination ($R^2$), where values closer to 0 indicate poor fit and values closer to 1 indicate perfect fit.

## Results

### Missing data and imputation

The data set contains missing data which is non-random, likely due to regulatory requirements, market demand, and client preferences. Regulatory requirements for chemotype testing vary over time and across jurisdictions but almost always require testing for the psychoactive cannabinoid THC and its acid form THCA. These testing requirements explain the low level of missing information (1.4%) for THC in the data (S1 Table, S1 Fig in S1 File). The growing popular interest in the therapeutic properties of (CBD) likewise explains its relatively low levels (37.78%) of missing data in the acid and neutral forms of CBD. Conversely, cannabichromene (CBC), cannabinol (CBN), tetrahydrocannabivarin (THCV), and cannabidivarin (CBDV) and their acid forms, show much higher levels of missing data since clients elect not to test for these poorly-understood and less-abundant cannabinoids.

To understand how the imputed values compared to the observed values, we plotted the kernel density estimates (Fig 1) for six of the seven cannabinoids. We excluded THC because of its few missing values. The graphs, particularly for the soft imputation in purple, becomes noisier when more data is missing. Therefore, for CBG and CBD (Fig 1A and 1B) the curves are much smoother given that these cannabinoids have only 30.27% and 37.78% of the data missing respectively (S1 Table in S1 File). However, for CBN and THCV (Fig 1D and 1E) the curves are jagged as they are missing 62% and 81.19% of the data respectively. The most extreme case is CBDV that lacks 96.6% of the data (Fig 1F).

The results of t-tests of each cannabinoid's observed distribution against each of its imputed distributions as a measure of goodness-of-fit is summarized in Table 2. Well-fit imputed values should have small test statistics and non-significant p-values while poorly-fit imputed values will have larger test statistics and significant p -values. The imputation methods generally struggled with the CBD, CBDV, and CBN values. At first glance this under-performance could be attributed to a lack of baseline information: CBDV had missing observations in 96.6% of the cases data and CBN was missing 60.3%. However, THCV (81.19%) and CBC (63.13%) were also missing large fractions of their observations but the imputation methods returned well-fit distributions. The iterative and k-nearest neighbor imputation method had significantly different distributions for four of the seven cannabinoids while the soft imputation method had well-fit distributions across all the cannabinoids.

To understand the relation of each of the seven cannabinoids against each other, we plotted the distribution and the relations among the observed data (Fig 2). The performance of each cannabinoid is portrayed by the kernel-density estimate of the (log-normalized) distribution of concentrations on the diagonal (Fig 2), while the upper triangle is a scatter plot of the

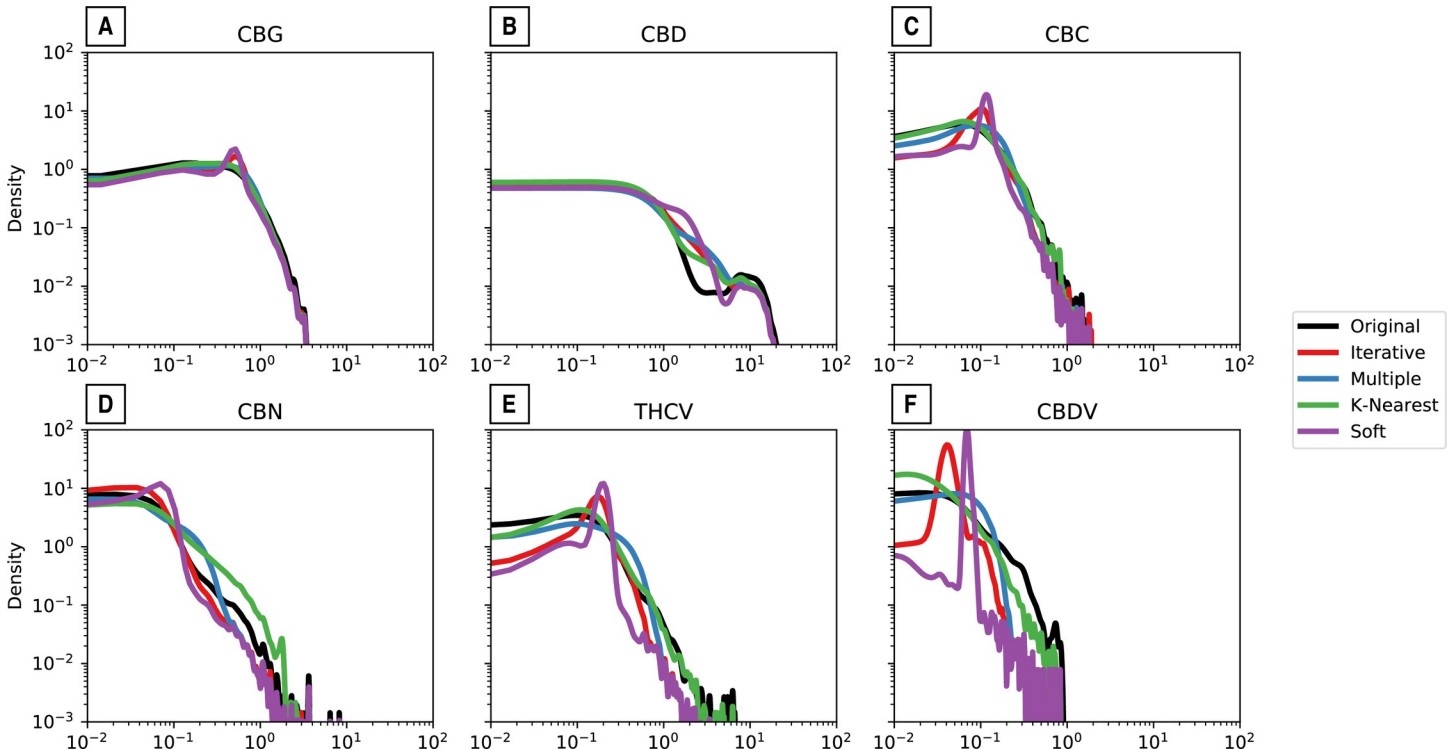

**Fig 1. Density plots of original data and imputed values for each of the cannabinoids.** The distribution of the observed values is in black, the iterative imputation in blue, the multiple imputation in red, the k-nearest neighbors in green, and the soft imputation in purple.

**Table 2. Statistics for observed vs. imputed cannabinoid concentrations.**

| Cannabinoid | Iterative | Multiple | K-Nearest | Soft |
|---|---|---|---|---|
| Cannabigerol (CBG) | -3.22 | -0.736 | 1.14 | -0.100 |
| | 0.00129** | 0.462 | 0.256 | 0.920 |
| Δ9-tetrahydrocannabinol (THC) | 0.749 | 0.605 | 0.633 | 0.315 |
| | 0.454 | 0.545 | 0.527 | 0.753 |
| Cannabidiol (CBD) | 4.45 | 5.77 | 3.51 | 1.76 |
| | <0.001*** | <0.001*** | <0.001*** | 0.077 |
| Cannabichromene (CBC) | 1.426 | 0.310 | 0.763 | 0.184 |
| | 0.154 | 0.757 | 0.445 | 0.854 |
| Cannabinol (CBN) | 5.62 | 2.98 | -22.3 | 0.219 |
| | <0.001*** | <0.003*** | <0.001*** | 0.826 |
| Tetrahydrocannabivarin (THCV) | 0.453 | -0.265 | -2.82 | 0.138 |
| | 0.650 | 0.791 | 0.005** | 0.890 |
| Cannabidivarin (CBDV) | 16.8 | 7.61 | 7.84 | 0.168 |
| | <0.001*** | <0.001*** | <0.001*** | 0.867 |

t-statistics (top) and p-values (bottom)

*$p < 0.05$

** $p < 0.01$

*** $p < 0.001$.

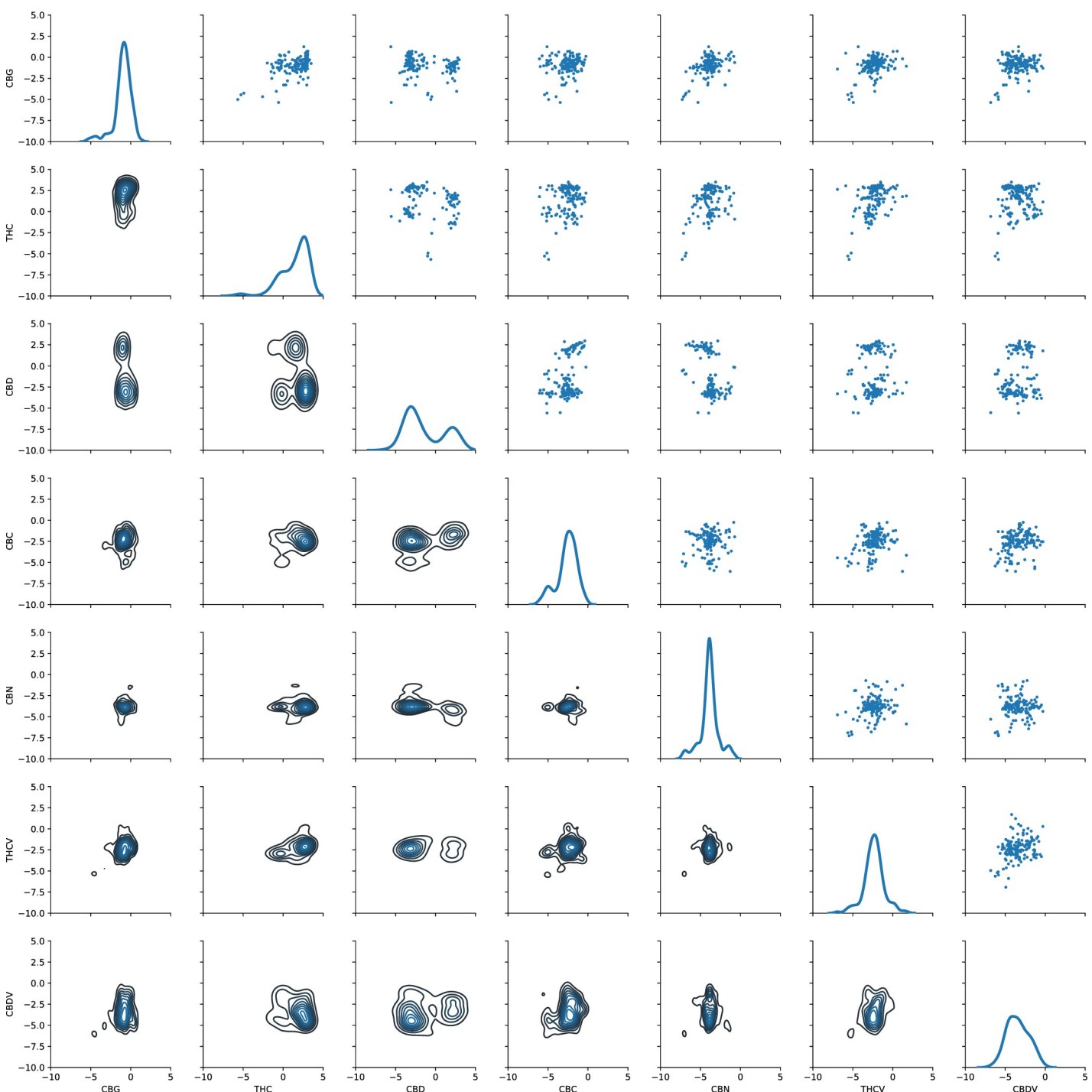

**Fig 2. Pair plot of the observed distributions of (log-scaled) cannabinoids.** Diagonal are kernel density estimates. Lower triangle are density plots and the upper triangle are scatter plots of the bivariate relationships.

relations between each of the seven cannabinoids against one another. The bottom triangle are density plots of the bivariate data distributions, which show distinct clusters on the common relation between each pair of cannabinoids.

The diagonal on Fig 2 shows that CBG is always around 0 given that it is found in very low quantities. On the contrary, the THC curve has two humps, a lower one around zero and a higher one in the positive side of the axis, showing that there are multiple individuals that have higher percentages of this cannabinoid. Similarly, the CBD curve also has two humps but much more conspicuous than the humps in the THC curve. The smoothness in the THC curve suggests that the data is continuous, different to CBD which has two clearly distinct groups. The bigger hump in CBD's curve suggests that there are more individuals that have little of the cannabinoid compared to the smaller hump with individuals that produce more CBD. Even though CBC also shows two humps, both suggest that most individuals have very little CBC. The humps for CBC, CBN, THCV, and CBDV have a maximum around zero establishing the lack of data for all of these cannabinoids.

The upper triangle on Fig 2 shows that CBG is found in very low concentrations compared to THC, CBD, and CBC. THC is always found in the positive side of the axis indicating both that the abundance of the data for this cannabinoid and that individuals produce it in larger amounts. Finding low levels of CBG is expected given that CBGA is the precursor molecule to which THCA, CBDA, and CBCA synthases act on to convert to THCA, CBDA, and CBCA respectively. Therefore, if these synthases are active, there should be little CBGA. This is exactly what our results show. Our results also show that the three cannabinoids can coexist, THC and CBD can be present together, as well as CBC and THC and CBC and CBD. However, CBC is always found in very low levels when compared to THC and CBD.

The density plots in the lower triangle (Fig 2) confirms that CBG is found in lower quantities compared to THC and CBD, and the lack of data for the rest of the cannabinoids. Yet the relationship between CBG and CBD is interesting, with two distinct clusters. Both clusters are centered around zero in the CBG (X) axis, showing the lack of this cannabinoid, but on the CBD (Y) axis one of the clusters shows its presence. The second cluster which is the bigger one (darker blue) suggests the lack of both CBD and CBG. The relationship between THC and CBD is also interesting with three clusters suggesting that most individuals have high THC and low CBD, but the other two clusters suggest the presence of (a) high-CBD individuals and (b) individuals that produce little of either cannabinoid (e.g., low THC and low CBD possibly hemp-type).

We plotted the two most well-known and biosynthetically related cannabinoids (THC and CBD) to better visualize their relationship (Fig 3). Four distinctive clusters are apparent in this visualization. Using the DBSCAN unsupervised clustering algorithm, we identified four substantive clusters of cultivars based on their distinctive combinations of CBD and THC concentrations (Fig 3). Cluster 0 (orange) contains high THC and low CBD cultivars, likely intended for recreational consumption. Cluster 1 (blue) contains low THC and high CBD cultivars, likely intended for medical consumption. Cluster 2 (pink) contains high THC and high CBD cultivars, likely intended for recreational consumption. Cluster 3 (green) contains low THC and low CBD cultivars, likely hemp intended for industrial use. The DBSCAN algorithm was not able to assign the cultivars in Cluster -1 to another cluster, so these remain outliers, but could theoretically be assigned to parent clusters through iteration, alternative clustering parameterization, or other clustering methods.

All of the imputation methods preserved substantive relationships like the negative THC vs. CBD correlation, and positive correlations for CBC vs. CBD, and CBN vs. CBC (Fig 4). The iterative imputation method introduced stronger correlations than existed in the original data between THC vs. CBDV, CBD vs. CBC, CBD vs. CBDV, and CBDV vs. CBC, (the red squares in the center of Fig 4B). The other imputation methods preserved similar pairwise correlation structures as were found in the original data. The imputation method that differs the most is the iterative imputation method (Fig 4B), which exaggerates the relationships and displays

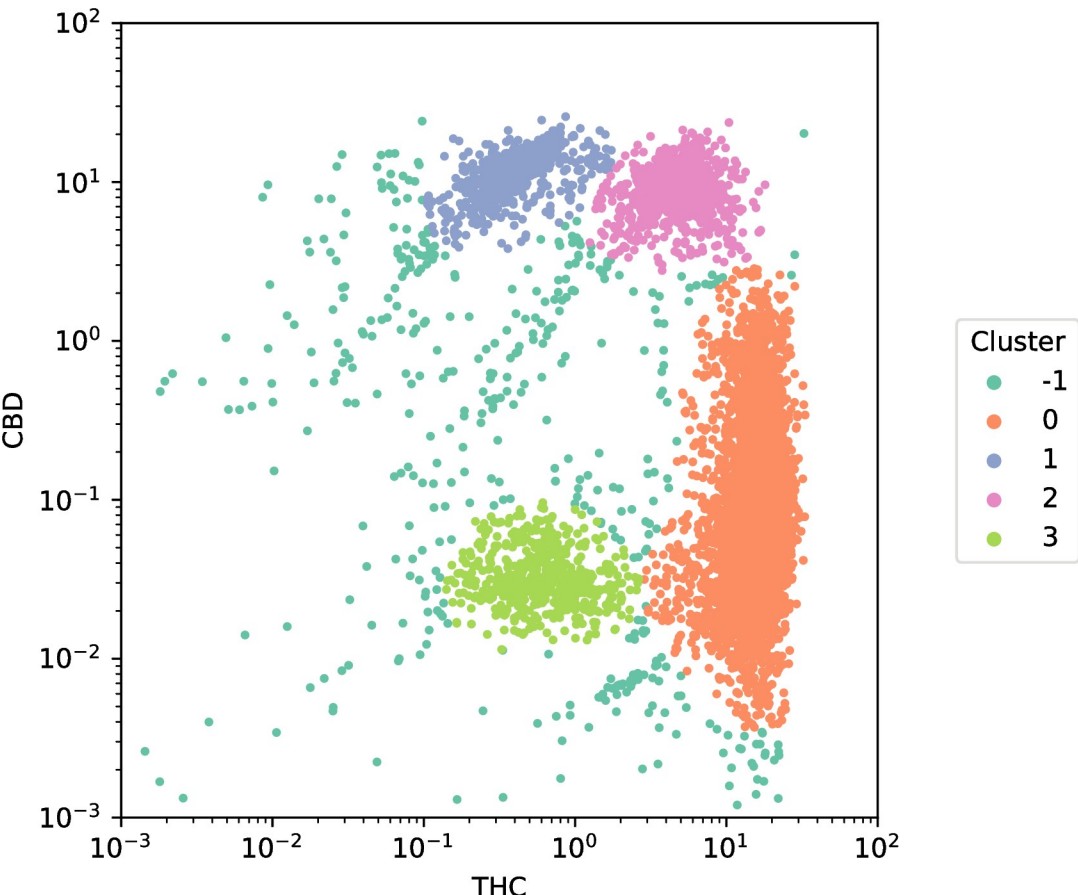

**Fig 3. Scatter of THC against CBD concentrations colored by cluster.** Four distinctive clusters are apparent in this visualization distinguished by different colors: **Cluster 0** in orange contains 8,270 cultivars and is characterized by high concentrations of THC and low concentrations of CBD, most likely intended for recreational use. **Clusters 1** in green contains 658 cultivars and is characterized by low concentrations of THC and high concentrations of CBD, most likely intended for medical consumption. **Cluster 2** in red contains 869 cultivars and is characterized by high concentrations of both THC and CBD, most likely intended for both recreational and medical use. **Cluster 3** in purple contains 536 cultivars and is characterized by low concentrations of both THC and CBD, most likely hemp-type of cultivars. Finally, **Cluster -1** in blue contains 433 cultivars that could not be assigned to other clusters using the DBSCAN algorithm, but were labeled using semi-supervised approaches like label propagation.

stronger correlations (red squares in the middle) than the original association. The K nearest neighbors (Fig 4D) is the one imputation method that weakens the relationships and the squares that are in orange in the original distribution (Fig 4A) are found in green.

Notice that the relationship between THC and CBD (purple squares) in the original relationship (Fig 4A) is maintained in the other imputation methods (Fig 4B–4E) oscillating between -0.5 and -0.6, suggesting a higher accuracy in predicting their behavior with any of the imputation methods. Additionally, this shows that the relationship between both cannabinoids can be more easily predicted compared to any of the other cannabinoids. This is expected, given their high potential concentrations and production from the same CBGA substrate. When THCA is produced there is less CBGA to produce CBDA and vice versa.

In order to understand how well the imputed values for the missing THC and CBD observations correspond to the clusters in Fig 3, we plotted the original distribution of data (in black) and the imputed values from each of the four imputation methods (Fig 5). The iterative

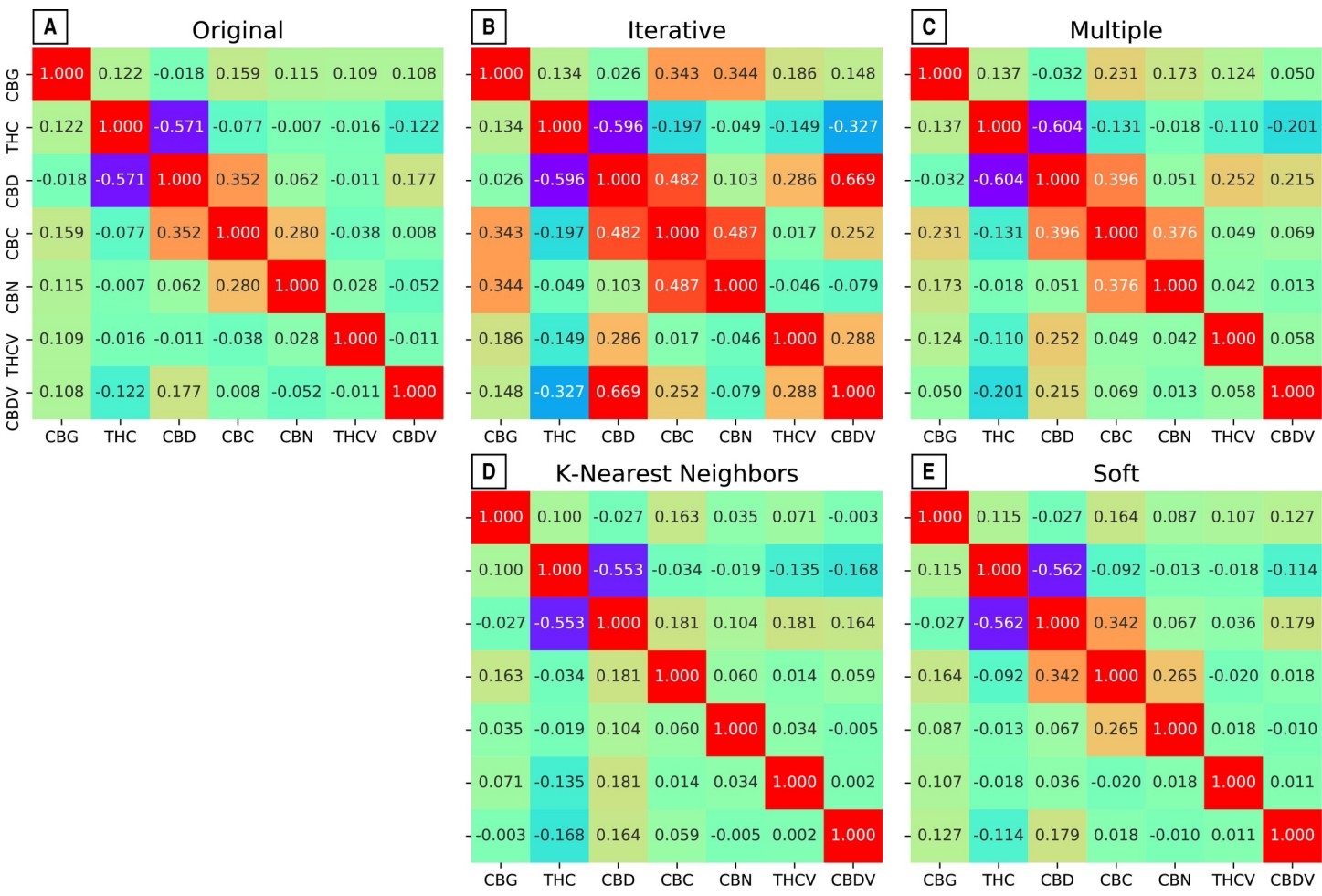

**Fig 4. Heatmap of the symmetric pairwise Pearson correlations.** Correlations were performed across the original data and the data from the four imputation methods.

(Fig 5A), multiple (Fig 5B), and soft (Fig 5D) imputation methods capture a quasi-linear relationship between CBD and THC (reflecting their competition for similar precursors). The K-nearest neighbors' method (Fig 5C) captures most of the patterns seen with the real data (Fig 3). All imputation methods capture clusters zero (high THC and low CBD), one (low THC and high CBD), and two (high THC and high CBD) from Fig 3 which comprise most of the data. However, all methods overlook the underlying clustered structure and miss the low-THC, low-CBD cluster (likely hemp, cluster 3 in green Fig 3). The failure of all methods to predict this cluster may be due to the few hemp varieties found in the data. Finally, even though K-nearest neighbor is the method that weakens the relationship between cannabinoids (Fig 4D) it is the one method that better overlays the original data for THC and CBD (Fig 5C).

## Statistical modeling

The values reported in Table 3 are the average R$^2$ from the five-fold cross validation for models predicting each cannabinoid and using data from each imputation method and different regression models. Given that model performance is evaluated using five-fold cross-validation to train a regression model on a subset of 80% of the data and test its performance on a held-out set of 20% of data. As a result, Table 3 reports 12 different scores for each cannabinoid:

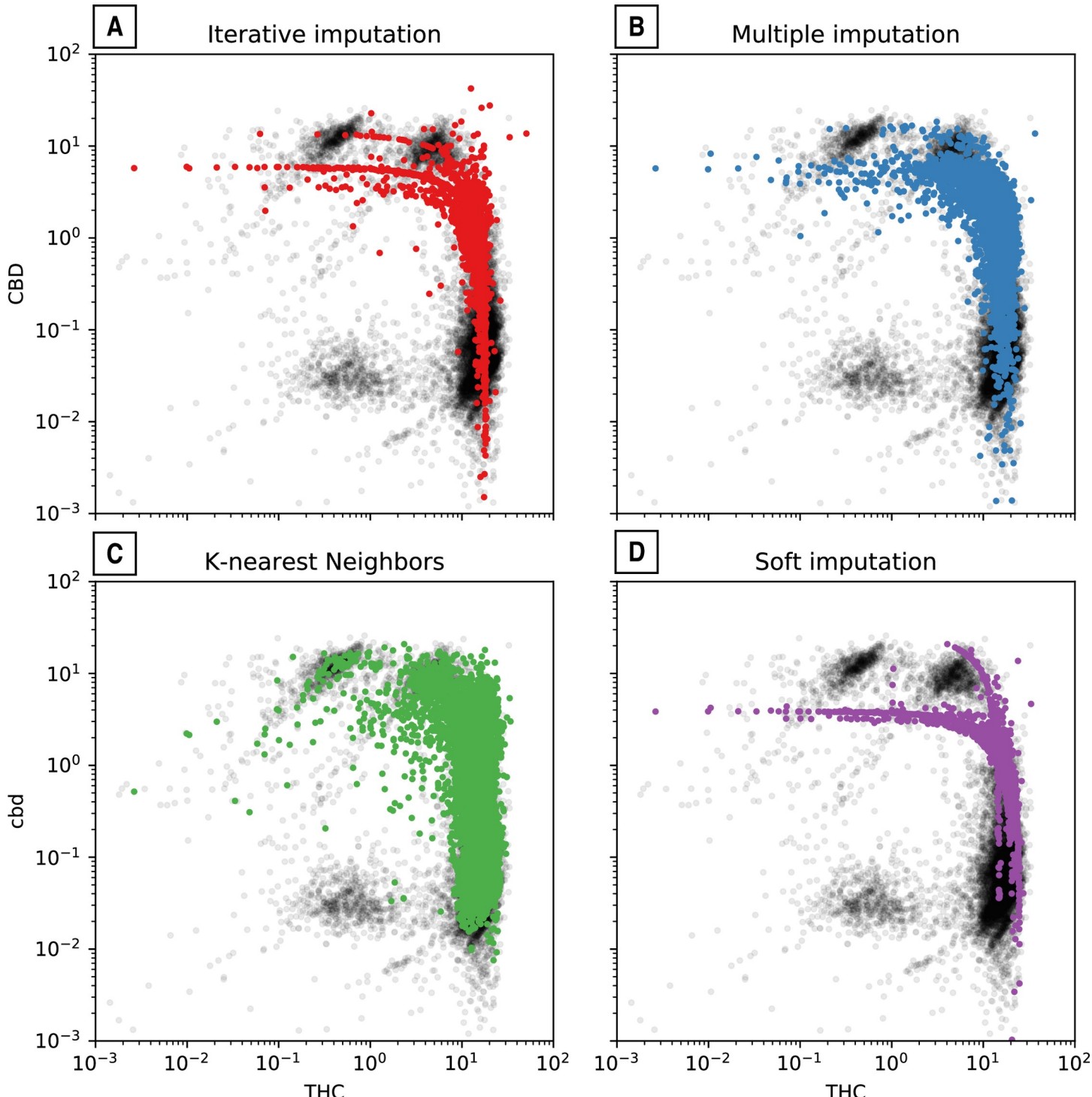

**Fig 5. Original vs. imputed values using four methods.** The 191 imputed values from each of the four methods, compared to the raw distribution on Fig 3.

three kinds of regression models on four kinds of imputation methods. The goal of this analysis is to identify if a particular combination of regression modeling and data imputation reliably models multiple kinds of cannabinoid abundance as a function of other cannabinoids. No

**Table 3. Average coefficients of determination ($R^2$) from a 5-fold cross-validation for different regression models.**

| Cannabinoid | K-Neighbors | | | | Linear | | | | Support Vector | | | |
|---|---|---|---|---|---|---|---|---|---|---|---|---|
| | I | M | K | S | I | M | K | S | I | M | K | S |
| CBG | 0.537 | 0.107 | 0.160 | 0.305 | 0.286 | 0.087 | 0.020 | 0.158 | **0.546** | 0.045 | 0.068 | 0.452 |
| THC | 0.634 | 0.304 | 0.353 | 0.597 | 0.403 | 0.403 | 0.323 | 0.357 | 0.526 | 0.443 | 0.408 | **0.688** |
| THCV | 0.309 | 0.155 | 0.068 | **0.628** | 0.091 | 0.029 | 0.041 | 0.062 | 0.011 | 0.012 | 0.008 | 0.065 |
| CBC | 0.395 | 0.169 | 0.047 | 0.236 | **0.459** | 0.321 | 0.053 | 0.150 | 0.367 | 0.241 | 0.077 | 0.361 |
| CBD | 0.811 | 0.515 | 0.515 | 0.747 | 0.528 | 0.454 | 0.385 | 0.403 | **0.815** | 0.562 | 0.542 | 0.627 |
| CBDV | 0.306 | 0.156 | 0.057 | 1.127 | **0.477** | 0.048 | 0.038 | 0.256 | 0.441 | 0.030 | 0.034 | 0.075 |
| CBN | **0.648** | 0.338 | 0.102 | 0.471 | 0.495 | 0.240 | 0.019 | 0.231 | 0.351 | 0.044 | 0.081 | 0.033 |

The three machine learning models -K-neighbors, linear, and support vector- predict the cannabinoid concentration using data from four different imputation methods: iterative, multiple, k-neighbors, and soft. The best-performing models are bolded. (The $R^2$ value for CBDV under K-Neighbors regression using soft imputation values exceeds 1 because of the mean squared error.)

regression model or imputation method is consistently better at predicting the cannabinoid concentration than others, but in some cases, particular cannabinoids can be very accurately predicted as is the case for CBD in the support vector model under the iterative imputation method ($R^2 = 0.815$). Additionally, some cannabinoids are more accurately predicted than others, which is the case of CBD or THC compared to CBC or CBDV.

## Dimensionality reduction

We projected each of the four imputation methods using the UMAP technique (Fig 6). Focusing on the four clusters, the UMAP projection clearly segregates high- and low-CBD and THC cultivars. This is an important confirmation that clustering based on the THC and CBD scores alone captured most of the substantive differences found using a non-linear and higher-dimensional technique. Low-THC strains (red and green) and high-THC strains (blue and purple) are likewise spatially proximate in three of the projections. Recall the DBSCAN clustering method used also failed to classify some observed points (blue in Fig 2, brown in Fig 6). Judging by their proximity to the other clusters in these projects, the unclassified points largely belong to the low-THC cultivars (red and green).

Cultivars whose CBD and or THC values were imputed are visualized in green (Fig 6). The iterative method (Fig 6A) classified most of these missing values in the high THC, low CBD (orange) cluster, but did identify a "new" grouping closer to the high THC, high CBD (blue) cluster. The projection of the multiple method (Fig 6B) returned at least two distinctive clusters with less overlap with the existing groupings. The projection of the k-nearest neighbors' method (Fig 6C) unsurprisingly shows a significant overlap with existing values, again mostly with the high-THC and low-CBD and the high-THC and high-CBD clusters. Finally, the values imputed from the soft imputation method (Fig 6D) have minimal overlap with any of the existing clusters and suggests a fifth group somewhere between the high-THC and low-CBD (orange) and the high-THC and high-CBD (blue) clusters.

## Discussion

In this study, we evaluated seven cannabinoids from 17,611 *Cannabis* samples, representing an unknown number of distinct varieties grown in four state-level markets within the United States. Due to the laws and regulations from the different states, we found that many cannabinoids were not tested, and this missing data is not random (S1 Table in S1 File). Therefore, our data consisted mostly of THC and CBD which are the widest-known cannabinoids. Even

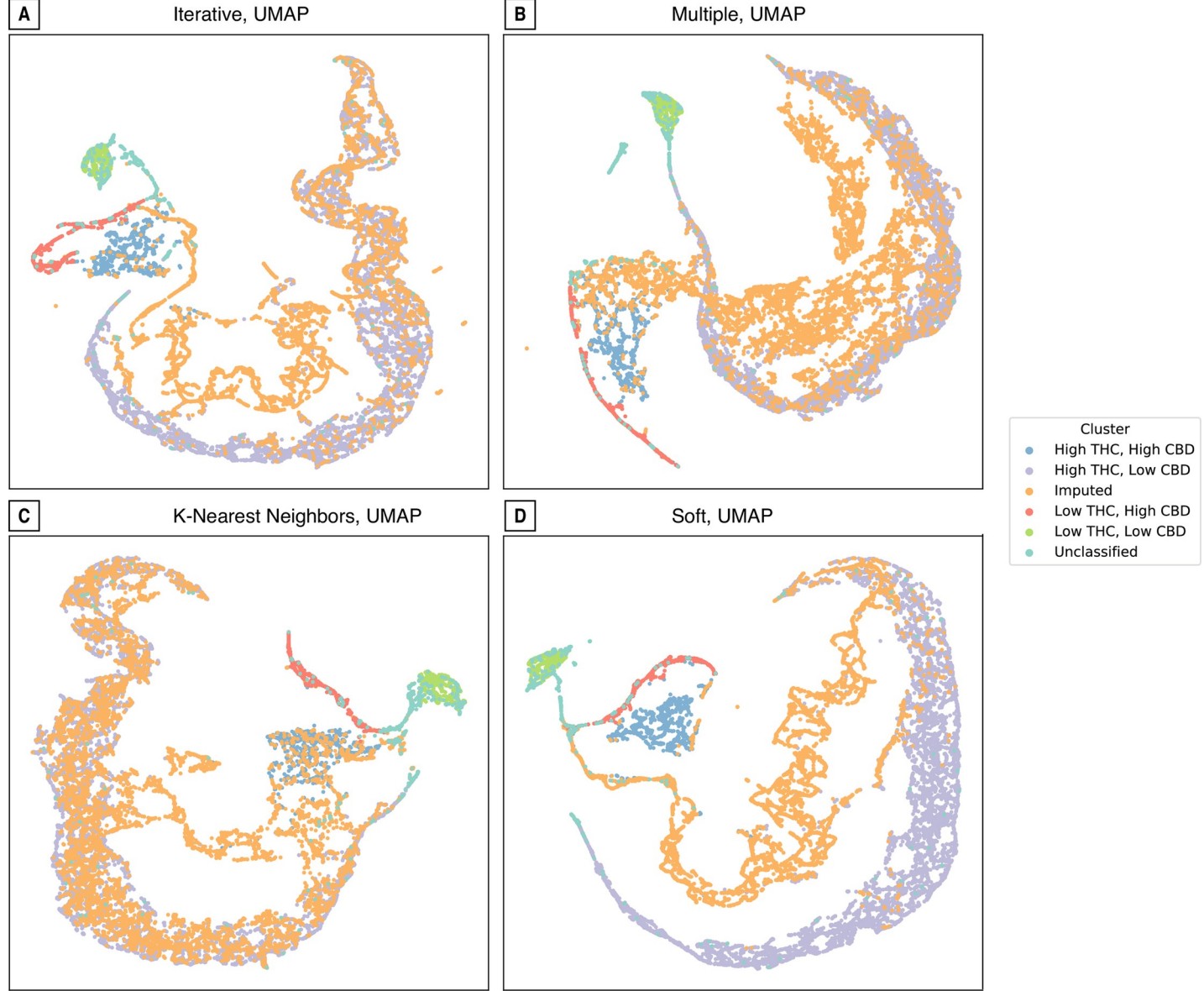

**Fig 6. Visualization of imputed data by reducing data with uniform manifold approximation and projection (UMAP).**

though our results met our expectations in clustering cultivars based on their chemotype, we found that some imputation methods are more robust at inferring missing data values, and due to the non-random absence of particular cannabinoids, their behavior could not be accurately imputed. This challenge in the imputation of missing values is particularly difficult for minor cannabinoids such as CBC, CBN, THCV, and CBDV. The results obtained are consistent with the assertion that there is no best method to impute these cannabinoids, and that subsets of the same dataset can behave differently with the same method (ie. Fig 4D, Fig 5C, Table 3).

The study reaffirms the misnaming of *Cannabis* varieties by the industry [18, 19], since strain identity cannot be predicted according to the clustering groups, even though the clusters are reflective of the chemotype (Fig 3 and S3 Fig in S1 File). The UMAP projection method

(Fig 6) segregates high- and low-CBD and THC cultivars, confirming that clustering based on these two cannabinoids captures most of the differences between varieties. Our results also agree with those who have shown that strain name is not indicative of potency or overall chemical composition [27]. Due to the naming issue, the clustering did not lead to visual separation of similar chemotypes to common groups. This misnomer issue in the *Cannabis* industry is further magnified by the fact that scientist must study the *Cannabis* produced by the federal government studies despite its inferiority in potency and diversity, and the fact that it does not reflect the products from the state medicinal and recreational markets [21, 22]. This is particularly problematic for medical patients who do not understand what they are consuming.

In order to improve the understanding of the *Cannabis* consumed for medical patients, chemotype testing must be made mandatory. However, testing facilities do not have standardized measurement protocols, cannabinoid analysis methods vary widely across laboratories [28], there are no institutional oversight to validate testing entities or their methodologies. In our study for example, it was hard to discern whether the presence of 0 (zero) for a given cannabinoid was non-detected or not-analyzed, which are two distinct assessments that should be evaluated differently. Given the lack of standardized laboratory methodologies and the lack of supervision of these testing facilities, differences in cannabinoid reporting is expected. However, despite these weaknesses and lack of standardization, some testing facilities do understand the importance and value of accurate chemotyping and see these chemotype tests as more than a service. Thanks to these type of facilities, important cannabinoid and terpenoid research has been achieved [21, 27, 29, 30].

The dimensionality reduction visualizations in Fig 6 cannot be compared with each other since they may be randomly transposed and the imputed values "displace" rather than "overlay" the original data. In spite of these limitations, a "good" projection will separate qualitatively distinct data into separate clusters while spatially preserving their latent similarity. Therefore, despite the different tradeoffs from all of our imputation methods, K-nearest (Fig 5C) displayed the best overlap with the original data. However, the coefficients of determination from the regression models (Table 3) did not show this particular method as the most accurate. Linear dimensionality reduction algorithms like PCA (did not perform as well as the non-linear methods like UMAP (Fig 6) and t-SNE. Finally, some cannabinoids are more accurately predicted than others, but no single imputation method or regression model provided consistent performance (Table 3, Fig 4). This could be due to the high amount of missing data (Table 1, S1 Fig in S1 File) but also suggests the need for incorporating theoretically-informed methods like flux balance analysis.

Because THC and CBD have attracted the most research and popular attention of all the cannabinoids and their synthases both compete for the same precursor CBGA, the connection between these compounds should reveal significant patterns. Our results show that despite the competition for the same precursor, all of these compounds can be present together. Our imputation results preserved associations such as the negative THC vs. CBD correlation, and positive correlations for CBC vs. CBD, and CBN vs. CBC (Fig 6). In fact, THC has a negative correlation with all other cannabinoids, and CBD and CBC always show a positive correlation (Fig 4). Additionally, CBC is always found in very low levels when compared to THC and CBD (Fig 2 upper half), which could indicate that CBCA synthase is not as good of a competitor for CBGA. The points near each of the axes where the percentage of that given cannabinoid is close to zero (Fig 2, upper half) suggest that those individual cultivars may either bear a synthase gene that is a bad competitor, or a truncated version of the enzyme [31]. These three enzymes could be classified as "sloppy enzymes" or "promiscuous" [32–34] particularly because in vitro they produce up to eight different compounds including each other's cannabinoids [35, 36]. Therefore, a third hypothesis could be that CBCA synthase is extremely sloppy

and produces more of the other cannabinoids, which would explain the larger amounts THC and CBD than CBC.

Some studies suggest that THC has been selected for and that varieties have been bred to increase in THC potency [37, 38]. Even though our results do not show this trend, the average THC from our results is much higher than that reported by others [37, 38]. THC production is probably a result of gene sequence variation [31], expression levels, and gene copy number variation [3], and there are multiple genes throughout the genome associated with its production [7, 39]. However, the expression of these genes could be due to environmental effects, and research suggests that the amount of THC is most likely due to cultivation conditions [27], which have not yet been measured. Additionally, the trends showing an increase on THC through time uses 20 years of data [37, 38], while we are only looking at eight years. This small snapshot of time probably underestimates the whole pattern of increase on THC, which suggests that breeders and growers have actively selected for this compound.

This work comes with limitations. First, we used the chemotypes from one company (Steep Hill) which may introduce endogenous biases due to the composition of clients and presence in markets with different regulations, and there are within and between lab differences in cannabinoid testing [28]. An argument can be made that the use of several labs will combine lab biases and their associated lab errors, that would tend to obfuscate some information in the case where analysis quality is poor in less experienced labs. It also can be expected that many labs have only collected more recent data or narrower group of cannabis cultivars as their origins are recent and the customer base may be small. This would tend to bias results towards the most popular cultivars from the last year or two. It was desired to examine a larger data set encompassing a longer time interval to include more cultivars and less local and temporal market preferences. Second, because some cannabinoids were systematically absent, we could not accurately infer their behavior. Third, the accuracy of the measurement method at low cannabinoid levels is poor, which may further complicate data collection particularly for those cannabinoids produced at lower quantities. Finally, future work could examine a broader profile of chemical profile contents including the missing cannabinoids and terpene data to allow for calculation of these inferences using ensemble effects which have become a popular analysis method [30]. Additionally, terpenoid compounds may be more information rich better in clustering the range of diverse *Cannabis* varieties [29, 30], but perhaps a better approach would be a combined analysis of both cannabinoids and terpenoids [12, 27] from multiple testing facilities.

## Supporting information

**S1 File.**
(DOCX)

## Acknowledgments

We thank two anonymous reviewers for their comments and suggestions.

## Author Contributions

**Conceptualization:** Daniela Vergara, Reggie Gaudino, Brian Keegan.

**Data curation:** Daniela Vergara, Reggie Gaudino, Thomas Blank, Brian Keegan.

**Formal analysis:** Daniela Vergara, Thomas Blank, Brian Keegan.

**Funding acquisition:** Daniela Vergara, Reggie Gaudino.

**Investigation:** Daniela Vergara.

**Methodology:** Daniela Vergara, Thomas Blank, Brian Keegan.

**Project administration:** Daniela Vergara, Reggie Gaudino.

**Resources:** Daniela Vergara, Reggie Gaudino.

**Software:** Brian Keegan.

**Supervision:** Daniela Vergara, Reggie Gaudino, Thomas Blank, Brian Keegan.

**Validation:** Reggie Gaudino, Thomas Blank, Brian Keegan.

**Visualization:** Daniela Vergara, Brian Keegan.

**Writing – original draft:** Daniela Vergara, Brian Keegan.

**Writing – review & editing:** Daniela Vergara, Reggie Gaudino, Thomas Blank, Brian Keegan.

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
