## [Decision Letter · Decision Letter 0]

9 Jun 2020

PONE-D-20-05885

Modeling cannabinoids from a large-scale sample of Cannabis sativa chemotypes

PLOS ONE

Dear Dr. Vergara,

Thank you for submitting your manuscript to PLOS ONE. After careful consideration, we feel that it has merit but does not fully meet PLOS ONE’s publication criteria as it currently stands. Therefore, we invite you to submit a revised version of the manuscript that addresses the points raised during the review process.

I would like to point you to the minor revisions outlined by reviewer #1. Please can you address these revisions. I look forward to receiving your revised manuscript.

We look forward to receiving your revised manuscript.

Kind regards,

Lucy J Troup, Ph.D

Academic Editor

PLOS ONE

Journal Requirements:

'This research was supported by donations to the Agricultural Genomics Foundation, to the University of Colorado Foundation gift fund 13401977-Fin8 to Professor Nolan C. Kane, and is part of the joint research agreement between the University of Colorado Boulder and Steep Hill Inc.'

We note that one or more of the authors are employed by a commercial company:Steep Hill Inc.

Additional Editor Comments (if provided):

Reviewers' comments:

Reviewer's Responses to Questions

**Comments to the Author**

1. Is the manuscript technically sound, and do the data support the conclusions?

Reviewer #1: Yes

Reviewer #2: Yes

2. Has the statistical analysis been performed appropriately and rigorously? 

Reviewer #1: Yes

Reviewer #2: Yes

3. Have the authors made all data underlying the findings in their manuscript fully available?

Reviewer #1: Yes

Reviewer #2: Yes

4. Is the manuscript presented in an intelligible fashion and written in standard English?

Reviewer #1: Yes

Reviewer #2: Yes

5. Review Comments to the Author

Reviewer #1: This is a very well written and interesting paper. I only have some very minor suggestions.

- The introduction is long. State clearly at the end the purpose of the study and the expectations.

- The discussion should be clearly structured. Start with the new findings and not with the repeat of the methods. What are the limitations of the study? Future studies? Conclusion paragraph.

- "Relationship" is not a statistical term. Please use correlation or association.

Reviewer #2: This work cluster cultivars and chemotypes together and show that cultivars cluster based on their chemotype. This work open a window for characterizing complex interactions in cannabinoid biosynthesis. Even the authors used a big number of cultivars and good bioinformatics tools, the work is very basic and not bring novel data or ideas.

I think this is a good solid work which can fit to Plose One policy of publication as the data is good and statistical analysis been performed appropriately.

6. PLOS authors have the option to publish the peer review history of their article (what does this mean?). If published, this will include your full peer review and any attached files.

Reviewer #1: No

Reviewer #2: No

---

## [Author Response · Author response to Decision Letter 0]

9 Jul 2020

Thank you for the time to read and improve our manuscript. We attached a response to reviewers document as required by the journal

---

## [Decision Letter · Decision Letter 1]

16 Jul 2020

Modeling cannabinoids from a large-scale sample of Cannabis sativa chemotypes

PONE-D-20-05885R1

Dear Dr. Vergara,

We’re pleased to inform you that your manuscript has been judged scientifically suitable for publication and will be formally accepted for publication once it meets all outstanding technical requirements.

Kind regards,

Lucy J Troup, Ph.D

Academic Editor

PLOS ONE

Additional Editor Comments (optional):

Reviewers' comments:

Reviewer's Responses to Questions

**Comments to the Author**

1. If the authors have adequately addressed your comments raised in a previous round of review and you feel that this manuscript is now acceptable for publication, you may indicate that here to bypass the “Comments to the Author” section, enter your conflict of interest statement in the “Confidential to Editor” section, and submit your "Accept" recommendation.

Reviewer #1: All comments have been addressed

Reviewer #2: All comments have been addressed

2. Is the manuscript technically sound, and do the data support the conclusions?

Reviewer #1: Yes

Reviewer #2: Yes

3. Has the statistical analysis been performed appropriately and rigorously? 

Reviewer #1: Yes

Reviewer #2: Yes

4. Have the authors made all data underlying the findings in their manuscript fully available?

Reviewer #1: Yes

Reviewer #2: Yes

5. Is the manuscript presented in an intelligible fashion and written in standard English?

Reviewer #1: Yes

Reviewer #2: Yes

6. Review Comments to the Author

Reviewer #1: The authors addressed all my comments very well. I have no further comments which might improve the quality of this paper. Thanks.

Reviewer #2: (No Response)

7. PLOS authors have the option to publish the peer review history of their article (what does this mean?). If published, this will include your full peer review and any attached files.

Reviewer #1: No

Reviewer #2: **Yes: **David Meiri

---

## [Editor Report · Acceptance letter]

21 Aug 2020

PONE-D-20-05885R1 

Modeling cannabinoids from a large-scale sample of Cannabis sativa chemotypes 

Dear Dr. Vergara:

I'm pleased to inform you that your manuscript has been deemed suitable for publication in PLOS ONE. Congratulations! Your manuscript is now with our production department. 

Kind regards, 

on behalf of

Dr. Lucy J Troup 

Academic Editor

PLOS ONE